# Do Interleukin-1 and Interleukin-6 Antagonists Hold Any Place in the Treatment of Atherosclerotic Cardiovascular Disease and Related Co-Morbidities? An Overview of Available Clinical Evidence

**DOI:** 10.3390/jcm12041302

**Published:** 2023-02-06

**Authors:** Athina Dimosiari, Dimitrios Patoulias, George D. Kitas, Theodoros Dimitroulas

**Affiliations:** 1Second Department of Internal Medicine, European Interbalkan Medical Center, 57001 Thessaloniki, Greece; 2Outpatient Department of Cardiometabolic Medicine, Second Department of Cardiology, General Hospital Hippokration, Aristotle University of Thessaloniki, 54642 Thessaloniki, Greece; 3Department of Rheumatology, Russells Hall Hospital, Dudley Group NHS Foundation Trust, Dudley DY1 2HQ, UK; 4Fourth Department of Internal Medicine, General Hospital Hippokration, Aristotle University of Thessaloniki, 54642 Thessaloniki, Greece

**Keywords:** cardiovascular disease, interleukin-1, interleukin-6, anakinra, canakinumab, tocilizumab

## Abstract

Cardiovascular disease (CVD) constitutes a real pandemic of the 21st century. According to data from the Centers for Disease Control and Prevention, one person dies every 34 min due to some form of CVD in the United States. Apart from the extremely high morbidity and mortality accompanying CVD, the economic burden seems to be unbearable even for developed countries in the Western World. The role of inflammation in the development and progression of CVD appears to be crucial, while, various inflammatory pathways, such as the Nod-like receptor protein 3 (NLRP3) inflammasome-interleukin (IL)-1/IL-6 pathway of the innate immunity, have attracted scientific interest during the last decade, as a potential treatment target in primary and/or secondary prevention of CVD. Whereas there is a significant amount of evidence, stemming mainly from observational studies, concerning the cardiovascular safety of IL-1 and IL-6 antagonists in patients with rheumatic diseases, evidence from relevant randomized controlled trials (RCTs) is rather scarce and conflicting, especially for patients without underlying rheumatic disease. In this review, we summarize and critically present the currently available evidence, both from RCTs and observational studies, concerning the place that IL-1 and IL-6 antagonists may hold in the treatment of CVD.

## 1. Introduction

Cardiovascular disease (CVD) is at the top of the leading causes of death in the United States of America (USA), as well as in Europe, despite growing knowledge of pathophysiology and intensive public health intervention strategies [1,2]. Globally, it is estimated that it accounts for 17.9 million deaths each year according to the World Health Organization (WHO). Almost 80% of these deaths are attributed to myocardial infarction and acute ischemic stroke, and one out of three patients are under 70 years old, while it is responsible for millions of disability-adjusted life years each year [1]. All this points out that CVD is a global epidemic in need of efficacious primary interventions and therapeutic solutions based on solid research data.

The term CVD is wide and mostly encompasses atherosclerosis, coronary artery disease (CAD), myocardial infraction (MI), ischemic stroke, and heart failure (HF), but also congenital and rheumatic heart disease, cardiomyopathies, pericardial and valvular diseases, peripheral vascular disease, pulmonary embolism, and arrhythmias. The range of these diseases along with the complicated interaction between environmental and genetic factors results in difficulties in recognizing the pathogenesis and etiology of CVD. Nevertheless, the hypothesis of vascular and myocardial inflammation seems common in most cases [3,4]. In the early 80s, findings of immune cells on the surface and within the atheromatic plaque, led to the theory of inflammation-mediated atherosclerosis, which was later confirmed by multiple pieces of evidence [5]. This area has attracted the interest of many researchers for years; however, the question that remains is whether targeting inflammation can be of clinical benefit in CVD.

Targeting inflammation is attractive, but also complicated. C-reactive protein (CRP) is an acute inflammatory protein, which is widely used to “quantify” inflammatory burden. There has been increasing evidence on the relationship between CRP and pro-inflammatory cytokines [6], suggesting that it is not only a marker of inflammation, but a regulator of inflammatory processes, as well. In addition, CRP is closely linked to CVD morbidity and mortality, as demonstrated by large studies and their meta-analyses in the field during the last two decades [7,8]. However, since CRP mediates various host responses, including complement pathway activation, apoptosis, phagocytosis, nitric oxide release, and cytokine production [6], it seems that it is a reliable marker of diagnosis and prognosis, but not a useful treatment target. Therefore, modern treatment strategies for CVD target inflammation, but utilize CRP for monitoring of treatment response.

## 2. Inflammation & Cardiovascular Disease

### 2.1. The Role of Interleukin-1

#### 2.1.1. Atherosclerosis

Lipid accumulation has been linked to low-grade arterial inflammation, since cholesterol microcrystals have been shown to activate inflammasome, resulting in the formation of active interleukin-1β (IL-1β), which further induces the production of other pro-inflammatory molecules, including interleukin-6 (IL-6) and the pro-inflammatory eicosanoid PGE2 [9]. This process partially contributes to the formation of atherosclerotic plaques, which are vulnerable, and finally rupture, resulting in an ischemic event [9]. Enhanced secretion of IL-1β correlates with increased secretion of reactive oxygen species (ROS), thus connecting inflammation with oxidative stress, and finally resulting in accelerated atherosclerosis and vascular damage [10]. In addition, IL-1β levels have been shown to correlate with the number of diseased vessels strongly and significantly among subjects with CAD, thus, pro-inflammatory cytokines have a crucial role not only for atherosclerosis pathogenesis but for the extent of atherosclerotic CVD, as well [11].

#### 2.1.2. Major Adverse Cardiovascular Events

IL-1β is counter-regulated by IL-1 receptor antagonist (IL-1RA), an endogenous inhibitor. Serum IL-1RA levels have been shown to correlate with increased risk positively and independently for incident CVD, after adjustment for possible confounders, including age, sex, anthropometric, metabolic, and lifestyle factors [12]. In addition, interleukin-33 (IL-33), a tissue-derived nuclear cytokine from the IL-1 family expressed in endothelial cells, epithelial cells, and fibroblast-like cells, and its unique receptor, ST2, have been shown to correlate with increased risk for all-cause death among individuals with established atherosclerotic CVD, especially in those who recently experienced an acute coronary syndrome [13]. IL-1 positive genotypes have been recently demonstrated to associate with increased risk for surrogate cardiovascular endpoints, including cardiovascular death, among individuals undergoing coronary angiography [14]. Of note, among subjects with chronic infections, such as those infected with human immunodeficiency virus (HIV), higher IL-1RA levels were shown to produce a 1.5-fold increase in the risk for MI, providing a rationale for the application of anti-inflammatory treatment strategies in order to decrease cardiovascular risk among patients with HIV [15].

#### 2.1.3. Peripheral Artery Disease

Peripheral artery disease (PAD) is another component of atherosclerotic CVD, linked to common co-morbidities such as hypertension, diabetes mellitus, and dyslipidemia. A former observational study demonstrated that, among subjects with various stages of PAD, circulating IL-1 levels were not different; however, IL-1RA levels significantly correlated with the PAD severity, from intermittent claudication to critical limb ischemia [16]. A higher inflammatory response, indicated by higher circulating levels of IL-1β, was found among patients with intermittent claudication undergoing maximal treadmill test, suggestive of its potential prognostic value for more advanced forms of the disease [17].

#### 2.1.4. Arterial Hypertension

Despite the undoubted role of IL-1 in the pathogenesis of atherosclerosis and CVD, a recent meta-analysis of observational studies involving 142,640 participants failed to demonstrate a significant association between IL-1 and the risk of developing hypertension in the general population [18]. Even when considering the association between IL-1 and arterial stiffness, a major cause of systolic hypertension in specific patient populations like the elderly and those suffering from chronic kidney disease, relevant evidence is contradictory [19,20].

### 2.2. The Role of Interleukin-6

#### 2.2.1. Atherosclerosis

Interleukin-6 (IL-6) is secreted following inflammasome activation and is implicated in the chronic inflammatory process of atherosclerosis [21]. Experimental data have been suggestive of a major role of IL-6 in vascular aging, since aged atherosclerotic aortas in mice have been shown to produce significantly greater concentrations of IL-6 and monocyte chemokines, compared to young aortas [22]. It has also been demonstrated that aged aortic smooth muscle cells (VSMC) exhibit a higher baseline secretion of IL-6 compared to young VSMC, along with upregulation of chemokines, like CCL2, adhesion molecules, and innate immune receptors, all of which substantially contribute to the development and exacerbation of atherosclerotic process [23]. Former data have documented that IL-6 is not only a significant contributor to the early onset of atherosclerosis but is also a marker of the greater extent of atherosclerotic lesions, involving other pro-inflammatory cytokines, like IL-1 and tumor necrosis factor alpha (TNF-α), as well [24].

#### 2.2.2. Major Adverse Cardiovascular Events

There is a significant amount of evidence concerning the interconnection between IL-6 and surrogate cardiovascular endpoints. Subjects with higher baseline circulating IL-6 levels have been shown to exhibit significantly greater risk for the development of CVD, strongly and positively correlating with blood pressure and total cholesterol levels, thus indicating a close relationship between low-grade inflammation and cardiovascular risk factors [25]. Especially in the elderly population, higher IL-6 levels have been demonstrated to contribute to an increased risk for cardiovascular death by 69%, also increasing the risk for all-cause mortality [26]. Among individuals with a recent acute coronary syndrome (ACS) event, it has been recently documented that higher baseline IL-6 levels significantly increase the risk for MACE by 29% and the risk for cardiovascular death by 55%, pointing out the significant prognostic value of IL-6 in the ACS setting [27]. Similarly, a close, significant, linear relationship between IL-6 levels and long-term risk of ischemic stroke has been demonstrated in the general population [28], while higher IL-6 levels have been also associated with significantly greater odds for poor functional outcomes after an acute ischemic stroke [29]. Therefore, evidence is suggestive of a significant, positive association between IL-6 and surrogate cardiovascular outcomes, including cardiovascular mortality, both in primary and secondary prevention. Of note, it appears that IL-6 is significantly more studied than IL-1 in clinical practice, regarding its potential applicability as a prognostic marker.

#### 2.2.3. Peripheral Artery Disease

IL-6 has been shown to correlate with the severity and the extent of PAD significantly and positively, having potential prognostic implications, along with imaging parameters [30,31]. However, other studies have found that IL-6 shows only a weak association with the risk for incident PAD in the general population, when other confounding factors are considered [32]. In addition, IL-6 does not seem to correlate with a higher rate of in-stent restenosis among subjects with PAD treated with endovascular therapy [33].

#### 2.2.4. Arterial Hypertension

Contrary to the absence of a relationship between IL-1β and the risk for incident hypertension, the same meta-analysis of observational studies demonstrated that higher baseline IL-6 levels are associated with a significant increase in the risk for incident hypertension by 51% [18]. In addition, it has also been documented that IL-6 levels are strongly and significantly correlated with systolic blood pressure levels [25], augmenting overall cardiovascular risk. Despite the fact that several inflammatory markers have been recently associated with increased risk for cardiovascular and all-cause mortality among hypertensive subjects, no such correlation has been shown neither for IL-6 nor for IL-1 [34].

## 3. IL-1 and IL-6 Antagonists for the Treatment of CVD

### 3.1. IL-1 Antagonists

The human recombinant IL-1 receptor antagonist (IL-1RA—anakinra) was designed to block the signal of IL-1α and IL-1β. It has a short half-life (4–6 h) allowing up and down titration, a good safety profile, an easy (although daily) route of administration via subcutaneous injection, and efficacy in the treatment of RA, juvenile arthritis, and cryopyrin-associated periodic syndromes (CAPS). In a similar way, canakinumab is a humanized monoclonal antibody selective against IL-1β and not IL-1α, best suited for long term use due to its monthly administration. Rilonacept is the last representative in this category of IL-1 blocking agents, acting as a chimeric decoying IL-1 recombinant receptor, binding IL-1α, IL-1β, and IL-1RA.


**Anakinra**


**Coronary artery disease** A previous, pilot randomized controlled trial (RCT) enrolling 10 patients with ST-segment elevation MI (STEMI), who were randomized to anakinra 100 mg/day subcutaneously for 14 days or placebo in a double-blind fashion, demonstrated that anakinra resulted in a significant improvement in both LV end-systolic and end-diastolic volume index, assessed both with echocardiography and cardiac magnetic resonance imaging (cMRI) [35]. Despite the major limitation of the small sample size, those results might be suggestive of a beneficial effect of anakinra on LV remodeling following acute MI, since adverse remodeling boosts morbidity and mortality in the short-term in that population [35]. Of note, no significant effect on LV ejection fraction or LV mass was demonstrated, while no difference between the two treatment groups was shown in terms of CRP and brain-type natriuretic peptide (BNP) levels [35].

In an early, non-randomized trial enrolling adults with RA, 23 of them received anakinra, and 19 received prednisolone for 30 days; anakinra was associated with a significant improvement in left ventricular (LV) systolic and diastolic function, along with a significant improvement in endothelial function, as assessed with flow-mediated dilation (FMD); however, no significant effect on pulse wave velocity (PWV), the “gold-standard” for arterial stiffness quantification, was documented [36].

In an acute, double-blind, crossover, RCT recruiting 60 patients with RA and pre-existing CAD, 20 patients with RA without baseline CAD, and 30 healthy controls, it was shown that a single dose of anakinra 100 mg resulted in a significant improvement in vascular function indices, including FMD, coronary flow reserve, systemic arterial compliance, and vascular resistance, and in a significant benefit on LV myocardial deformation, twisting, and untwisting [37]. Of note, LV function improvement was greater among patients with prior CAD, possibly correlating with a greater reduction in oxidative stress markers and improvement in vascular function, compared to those without such a history [37].

Another previous, relevant RCT, the MRC-ILA Heart Study, recruited 182 patients with non-ST elevation ACS, who were randomized 1:1 to receive either anakinra 100 mg once daily or a placebo, for 14 days [38]. The study demonstrated significant suppression of inflammatory markers with anakinra treatment, but the short duration of therapy did not allow for therapeutic outcomes to unveil [38]. However, in terms of “hard” cardiovascular endpoints, there was no significant difference in MACE incidence at 30 and 90 days, while, remarkably, there was a significant increase in the one-year risk for MACE with anakinra treatment [38].

In the field of coronary artery disease, another RCT, the VCUART3 (Virginia Commonwealth University Anakinra Remodeling Trial 3) trial, showed a significant reduction in inflammatory burden among 99 STEMI patients with the use of anakinra, no difference between standard and high dose regimens (100 mg once or twice daily), establishing the knowledge that IL-1 blockage suppresses the system in inflammatory response among those patients [39]. In terms of LV remodeling, however, no significant effect on LV function was shown [39]. Regarding surrogate cardiovascular endpoints, it was documented that anakinra compared to placebo did not have any significant effect on mortality; however, it significantly decreased the risk for incident HF, including in inpatient and outpatient settings [39].

**Heart failure** HF as a condition itself is a state of chronic inflammation in which cytokine expression and excretion are stimulated as a response to hypoxic stress, anaerobic glycolysis, and impaired calcium homeostasis [40,41]. In this field, the recently published REDHART (Recently Decompensated Heart Failure Anakinra Response Trial), another RCT, studied the effect of anakinra 100 mg once daily for 14 days or 12 weeks versus placebo on exercise capacity in 60 patients with recently decompensated systolic HF, showing an improvement in peak aerobic exercise capacity after 12 weeks of treatment, as perceived by the patients; however, without any improvement in cardiopulmonary exercise testing main variables, including peak VO2, and the VE/VCO2 slope [42]. Of course, as confirmed in the initially published report of the REDHART trial, no significant effect of anakinra on surrogate cardiovascular endpoints was shown [43]. Notably, those patients with a LV ejection fraction (LVEF) lower than 35% experienced a significant improvement in peak VO2 with 12 weeks of treatment with anakinra; however, such a trend was not confirmed with shorter duration treatment with anakinra [43]. On the other hand, application of the same treatment protocol in 31 obese patients with HF with preserved LVEF demonstrated that 12-week treatment with anakinra compared to placebo failed to improve peak VO2 or LVEF, despite improving perceived functional capacity and suppressing systemic inflammation, as reflected by CRP levels [44].

**Pulmonary hypertension** Pulmonary arterial hypertension (PAH) is another increasingly recognized form of CVD, affecting millions of patients worldwide, with its pathophysiology encompassing among others, systemic inflammation, maladaptive cytokine signaling, and development of right ventricular failure and HF, regardless of underlying etiology [45]. PAH is a promising new field to explore the use of interleukin-blocking agents [45]. In a recently published, single-group, open-label phase Ib/II pilot study including six patients with PAH and symptomatic right ventricular (RV) failure, anakinra treatment 100 mg once daily for 14 days led to a significant decrease in high-sensitivity CRP (hsCRP) levels, without any improvement in IL-6 levels, peak VO2 consumption or echocardiographic parameters of RV systolic function, such as fractional area change or tricuspid annular plane systolic excursion [46]. Despite being safe, the small sample size and study design did not permit drawing further conclusions regarding the true efficacy of anakinra for the treatment of PAH and related RV failure [46].

In the field of atherosclerosis and atherothrombosis management, current treatment modalities are primarily designed to target cholesterol reduction. The emerging data on the critical role of inflammation in the development of CAD and its components, along with data supporting the downregulation of inflammatory biomarkers with the use of statins, urged researchers to test other anti-inflammatory drug classes [47,48].

To sum up, current evidence concerning the cardiovascular efficacy of anakinra is rather conflicting; available studies confirm the anti-inflammatory effect of this drug, as quantified with the use of inflammatory markers, however, this effect is not translated into a clinically meaningful benefit.


**Canakinumab**


**Atherosclerotic CVD** The hallmark CANTOS (Canakinumab Anti-Inflammatory Thrombosis Outcome Study) trial opted to test the effect of canakinumab on the total number of CVD events (first or recurrent), in a randomized, double-blinded, placebo-controlled trial of 10,061 patients with established atherosclerotic CVD (past history of MI) and residual inflammatory risk [49]. Enrolled participants were randomized to receive various doses of canakinumab (50 mg, 150 mg, and 300 mg administered subcutaneously every three months) or placebo and were followed up for a median of 3.7 years [49]. Canakinumab 150 mg resulted in a significant decrease in the risk for the primary composite endpoint (nonfatal MI, nonfatal stroke, or cardiovascular death) compared to placebo by 15%, while canakinumab 50 mg and 300 mg failed to show any superiority compared to placebo [49]. Canakinumab 150 mg cardiovascular efficacy was mainly driven by its beneficial effects on CAD components, since it provided a significant decrease in the risk for MI by 24%, for coronary revascularization by 32%, and for hospitalization due to unstable angina by 36% [49]. In a subsequent, subgroup analysis, it was shown that canakinumab at all three doses resulted in a significant decrease in the risk for MACE among those randomized subjects with prior MI and residual inflammatory risk [50]. Another subgroup analysis of the CANTOS trial documented a dose-dependent reduction in the risk for HF hospitalization with all doses of canakinumab, compared to placebo [51]. **Peripheral artery disease** Likewise, some insights into the possible benefits of IL-1 blockage in PAD were provided by a small RCT recruiting 38 symptomatic PAD patients, which demonstrated that 12-month treatment with canakinumab 150 mg compared to placebo resulted in a significant decrease in inflammatory markers, such as hsCRP and IL-6, along with a significant improvement in walking distance and exercise capacity; however, canakinumab failed to ameliorate superficial femoral artery atherosclerotic burden [37]. Those data were strongly suggestive that among patients with established CVD, anti-inflammatory treatment with canakinumab is insufficient to induce regression of the plaque in the microvasculature [52].

It should be highlighted that IL-1 might also be implicated in the pathogenesis of type 2 diabetes [53]. The use of anakinra has been shown in a small RCT to improve glycemic control and systemic inflammation in patients with type 2 diabetes [54]. Consequently, anakinra might be of benefit to this high-risk patient population [54]. At the same time, canakinumab also seems to be safe for patients with type 2 diabetes [55]. This agent led to significant suppression of systemic inflammation in a group of patients with atherosclerotic disease and either type 2 diabetes or impaired glucose tolerance after 12 months of treatment, with an additional benefit of significant lipoprotein (a) reduction, however, without any significant effect on vascular function and structure, at the level of carotid arteries and aorta [56]. Of note, no significant effect on glycemic control and insulin resistance was documented with the use of canakinumab versus placebo [56].

In conclusion, evidence mainly generated by the CANTOS trial suggests that an intermediate dose of canakinumab can provide a cardiovascular benefit for patients with pre-existing CVD; however, additional, confirmatory evidence since the original publication of CANTOS is still lacking.


**Rilonacept**


**Recurrent pericarditis** Last but not least, rilonacept, an IL-1α and IL-1β cytokine trap, has also been shown in a recently published, phase 3 RCT, to be beneficial for patients with recurrent pericarditis, resulting in a significantly lower risk for pericarditis recurrence, compared to placebo [57]. However, rilonacept has not been tested so far in patients with established atherosclerotic CVD, for the assessment of its effects on endothelial or vascular function, or crude cardiovascular outcomes [57].

In conclusion, rilonacept is an effective treatment option in patients with recurrent pericarditis; however, its efficacy in other forms of CVD has not been tested.

An overview of the relevant RCTs is provided in Table 1.

### 3.2. IL-6 Antagonists

IL-6 is a multifunctional pro-inflammatory cytokine that has been proven to take part in atherosclerosis development, destabilizing the atherosclerotic plaque, substantially contributing to acute myocardial ischemia, and inhibiting reperfusion [58]. Its’ major role in CVD, and especially CAD, mainly involved endothelial dysfunction and overexpression of other pro-inflammatory markers in the coronary endothelial cells. On the other hand, although prompt coronary intervention in ACS reduces infarct size and ensures myocardial viability, it is also established that prolonged inflammation through upregulation of IL-6 is associated with adverse LV remodeling and, thus, worse prognosis [59].


**Tocilizumab**


**Primary prevention** In a large RCT including 3080 seropositive active RA patients, tocilizumab at a dose of 8 mg/kg/month proved to have a similar cardiovascular safety profile compared to etanercept at 50 mg/week, since no statistical difference across surrogate cardiovascular endpoints was shown after a median follow-up period of 3.2 years [60]. Another relevant RCT, the MEASURE trial, in a total of 132 patients with RA, showed that treatment with tocilizumab plus methotrexate compared to placebo plus methotrexate led to a significant increase in total cholesterol, low-density lipoprotein-cholesterol (LDL-C) and triglyceride levels by week 12, while no difference in high-density lipoprotein-cholesterol (HDL-C) levels and small, dense LDL was shown [61]. Despite the significant suppression of systemic inflammation, as quantified by the measurement of various inflammatory markers (including CRP, fibrinogen, and paraoxonase), tocilizumab failed to improve vascular function, as assessed with arterial stiffness, compared to placebo [61]. Despite the global increase in cholesterol levels with tocilizumab treatment, previously published real-world data have suggested that the cardiovascular events observed in RA patients under tocilizumab treatment are in the range of what is expected due to age, co-morbidities and disease activity, regardless of tocilizumab treatment [62].

In a nationwide cohort study including 1584 patients with RA not responsive to TNF-α inhibitors, second-line treatment with tocilizumab compared to rituximab resulted in a significant decrease in the risk for MI by 88% and for a MACE by 59% [63]. A recent network meta-analysis synthesizing data from both RCTs and observational studies enrolling RA patients has demonstrated that tocilizumab has a well-demarcated cardiovascular safety profile compared to other RA treatment options, while it might decrease the risk for MI compared to abatacept [64]. Another relevant meta-analysis showed that tocilizumab might decrease the risk for MACE in patients with RA, compared to tumor necrosis factor inhibitors, with the result being marginally non-significant [65].

Real-world data are also suggestive of a beneficial effect of tocilizumab on cardiac function among subjects with RA without pre-existing CVD. In a former observational study enrolling 20 female patients with RA and 20 age-matched healthy controls it was shown that, after 52 weeks of tocilizumab treatment, RA patients experienced a significant increase in LVEF and a significant decrease in LV mass index, along with normalization of baseline eccentric LV hypertrophy [66]. Of note, the percentage change in LV mass index strongly correlated with the percentage change in disease activity, as assessed with the Simplified Disease Activity Index [66]. In another observational study recruiting 70 patients with RA, free of CVD at baseline, it was documented that 24-week treatment with tocilizumab led to a significant decrease in N-terminal pro-brain natriuretic peptide (NT-proBNP) levels, a change that was shown to strongly correlate with the observed change in disease activity [67]. Of course, there are also some other observational data demonstrating that different biologics do not exert a differential effect on cardiac damage biomarkers among patients with RA, pointing out the need for further, more focused, research in this field [68].

**Coronary artery disease** In the field of CAD, the detrimental effects of prolonged inflammation after an ACS are connected to increased IL-6 levels and complement activation [69]. Although tocilizumab administration does directly not impair complement activation, it suppresses the excretion of the highly inflammatory anaphylatoxins C5aR1, C5aR2, and C3aR among patients with non-ST segment elevation MI [69]. In another relevant RCT enrolling patients with non-ST segment elevation MI, a single dose of tocilizumab prior to coronary angiography, compared to placebo, resulted in a significant increase in citrullinated histone 3 (H3Cit) levels, which are associated with the formation of neutrophil extracellular traps (NETs), carrying inflammatory and thrombotic properties, posing questions regarding its true anti-inflammatory efficacy in the acute setting [70]. Sub-analysis of the above-mentioned trial showed that a single dose of tocilizumab compared to a placebo did not have a significant impact on coronary flow reserve, both during hospitalization and six months after the ACS event [71].

Another trial recruiting a total of 199 patients documented that, when tocilizumab is administered within the first 6 h after onset of symptoms in patients with ST-segment elevation MI, it produces a significant increase in myocardial salvage index, quantified with cardiac magnetic resonance imaging, compared to placebo, although no significant effect on final infarct size was shown [72].

Of interest, as shown in the recently published IMICA trial [73], which enrolled 80 comatose patients with out-of-hospital cardiac arrest, a single infusion of tocilizumab (8 mg/kg) compared to placebo, resulted in significant suppression of systemic inflammation, along with a significant reduction in myocardial injury and myocardial stress biomarkers, including troponin T, creatine kinase myocardial band and NT-pro-BNP. No significant difference in terms of cardiovascular safety or efficacy was demonstrated; however, patients treated with tocilizumab experienced a significant increase in the risk for initiation of renal replacement therapy during intensive care unit hospitalization, although renal function returned to baseline levels in all survivors that were finally discharged [73].

In conclusion, tocilizumab has a potent anti-inflammatory effect among subjects with high residual inflammatory burden, such as those suffering from RA; however, that effect is not translated into clinical benefit across crude cardiovascular endpoints of interest, including mortality.


**Sarilumab**


Data concerning the cardiovascular safety and efficacy of sarilumab, a human monoclonal antibody that binds membrane-bound and soluble IL-6 receptor-α to inhibit IL-6 signaling, are rather scarce, although a post-hoc analysis of a previously published trial confirmed the superiority of sarilumab compared to adalimumab in terms of systemic inflammation suppression, along with a significantly greater reduction in lipoprotein (a) levels, an established cardiovascular risk factor [74].


**Ziltivekimab**


**Primary prevention** Finally, cardiovascular safety and efficacy of ziltivekimab, a fully human monoclonal antibody directed against the IL-6 ligand, was tested in a phase 2 RCT published in 2021, the RESCUE trial [75]. In a total of 66 subjects with moderate to severe chronic kidney disease and residual inflammatory risk, treatment with ziltivekimab 7.5 mg, 15 mg, or 30 mg every four weeks, up to 24 weeks, compared to placebo, resulted in a significant decrease in hsCRP levels in a dose-dependent manner, which remained stable over treatment period [75]. The dose-dependent decrease was also observed for other inflammatory markers, including fibrinogen, serum amyloid A, haptoglobin, secretory phospholipase A2, and lipoprotein (a) [75]. No major safety issue arose, while no significant effect on total cholesterol to HDL-cholesterol levels was shown [75]. Again, according to this evidence, the presence of clinical benefit for patients with or at high risk for CVD cannot be deduced.

An overview of the relevant RCTs is provided in Table 2.

## 4. Future Perspectives and Concluding Remarks

Targeting of the Nod-like receptor protein 3 (NLRP3) inflammasome-IL-1/IL-6 pathway of innate immunity for the treatment of CVD has attracted major scientific interest during the last decade [76]. Despite the fact that the inflammatory hypothesis of atherosclerosis is conflicted with additional, modifiable, or not, cardiovascular risk factors [77], it is undeniable that the NLRP3 inflammasome activation contributes to vascular inflammation, leading to the establishment and progression of atherosclerosis and CVD [78]. Targeting pro-atherogenic and pro-inflammatory cytokines, such as IL-1 and IL-6, appears to be a reasonable treatment strategy for CVD [79]. The only “anti-inflammatory” treatment that has been shown to provide significant cardiovascular benefits, mainly in the field of secondary prevention of CVD, is colchicine [80,81], while others, such as low-dose methotrexate, have completely failed in this direction [82].

Therefore, inflammatory cascade appears to be an interesting treatment target in CVD. Nowadays, despite early intervention treatment protocols, precise revascularization techniques, and broad use of drug classes such as renin-angiotensin-aldosterone system blockers, statins, and proprotein convertase subtilisin/kexin type 9 inhibitors, the incidence of CVD remains unacceptably high. Thus, besides preventive strategies, it is also of utmost importance to identify novel, efficacious agents with an acceptable safety profile to improve treatment outcomes.

It is well understood and believed that blocking atherogenesis’ inflammatory pathways might provide beneficial, cardiovascular effects. However, except for the CANTOS trial, the rest available data are rather limited and, most importantly, conflicting. In addition, in the era of coronavirus disease-2019 and its strong relationship with hyper-inflammation, it still has to be determined whether these agents can prevent short- and long-term cardiovascular complications of the disease, based on their widespread use in the acute setting of severe disease [83,84,85].

In addition, concerning other cardiovascular risk markers, mainly lipid profile parameters, IL-1 antagonists, and mainly canakinumab, have been shown to have a neutral effect on low-density lipoprotein cholesterol (LDL-C) and high-density lipoprotein cholesterol (HDL-C), however increasing triglyceride levels [86], while IL-6 antagonists have been demonstrated to produce a significant increase in LDL-C, HDL-C, and triglyceride levels, prompting initiation of lipid-lowering agents (statins) early after the onset of treatment [87,88].

Upon to current evidence, and despite the significant progress in the understanding of pathophysiologic mechanisms that are affected by IL-1 and IL-6 (Figure 1), initiation of either IL-1 or IL-6 antagonists for the primary or secondary prevention of CVD should not be recommended, except for patients with underlying rheumatic disease, with clear indication for anti-cytokine treatment. Well-designed trials, ideally with a head-to-head comparison of anti-inflammatory agents, are warranted to provide definitive answers (Figure 2).

## Figures and Tables

**Figure 1 jcm-12-01302-f001:**
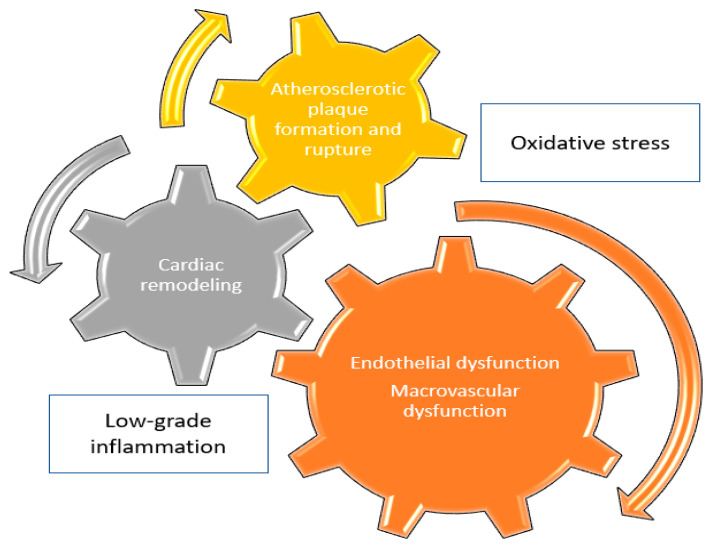
Major pathophysiologic mechanisms implicated into CVD pathogenesis and affected by IL-1 and IL-6.

**Figure 2 jcm-12-01302-f002:**
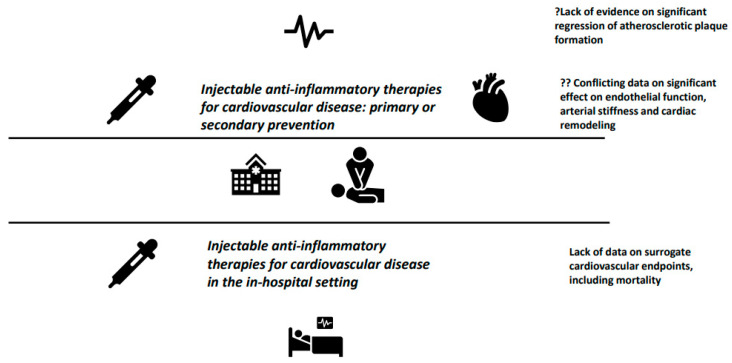
Injectable IL-1 and IL-6 antagonists for the primary and secondary prevention of major forms of CVD, as well as treatment of CVD in the in-hospital setting: conflicting pathophysiologic data and lack of high-quality clinical trials.

**Table 1 jcm-12-01302-t001:** Overview of randomized controlled trials assessing the cardiovascular effects of IL-1 antagonists.

Study	Type of Study	Treatment Duration	Study Population	Utilized IL-1 Antagonist	Comparator	Main Findings
Anakinra						
Abbate et al., 2013 [35]	Randomized controlled trial	14 days	10 patients with ST-segment elevation acute myocardial infarction	Anakinra 100 mg once daily	Placebo	Anakinra compared to placebo resulted in: Significant improvement in LV end-systolic and end-diastolic volume index No significant effect on LV ejection fraction or LV mass No significant effect on C-reactive protein (CRP) levels No significant effect on brain-type natriuretic peptide (BNP)
Ikonomidis et al., 2014 [37]	Randomized controlled trial	3 h	60 patients with RA and coronary artery disease (CAD), 20 patients with RA without CAD and 30 healthy controls	Anakinra 100 mg as a single dose		Anakinra compared to placebo resulted in: Significant improvement in vascular function indices Significant improvement in LV myocardial deformation, twisting and untwisting Significant decrease in oxidative stress markers * Patients with RA + CAD experienced greater benefits compared to those with RA without CAD
Morton et al., 2015 [38]	Randomized controlled trial	14 days	182 patients with non-ST elevation acute coronary syndrome	Anakinra 100 mg once daily	Placebo	Anakinra compared to placebo resulted in: Significant reduction in inflammatory markers No significant effect on myocardial injury biomarkers No significant effect on the risk for major adverse cardiovascular event at 30 or 90 days Significant increase in the risk for incident major adverse cardiovascular event at 1 year of follow-up
Abbate et al., 2020 [39]	Randomized controlled trial	14 days	99 patients with non-ST segment elevation acute myocardial infarction	Anakinra 100 mg once or twice daily	Placebo	Anakinra compared to placebo resulted in: Significant decrease in inflammatory markers No significant effect on LV remodeling No significant effect on the risk for death or stroke Significant decrease in the risk for incident heart failure
Van Tassell et al., 2017 [43] & Mihalick et al., 2022 [42]	Randomized controlled trial	2–12 weeks	60 patients with recently decompensated heart failure	Anakinra 100 mg once daily for 2 weeks or for 12 weeks	Placebo	Anakinra compared to placebo resulted in: Significant improvement in patients’ perception of dyspnea and exertion at submaximal exercise Significant reduction in inflammatory markers No effect on LV systolic function No effect on the risk for “hard” cardiovascular endpoints * In the 12-week anakinra treatment arm, patients with a LV ejection fraction (LVEF) lower than 35% experienced a significant improvement in peak VO_2_ during cardiopulmonary exercise testing
Van Tassell et al., 2018 [44]	Randomized controlled trial	12 weeks	31 patients with heart failure with preserved left ventricular ejection fraction	Anakinra 100 mg once daily for 12 weeks	Placebo	Anakinra compared to placebo resulted in: Significant improvement in inflammatory markers Significant improvement in perceived functional capacity No effect on peak VO_2_ No effect on left ventricular ejection fraction
Canakinumab						
Ridker et al., 2017 [49]	Randomized controlled trial	48 months	10,061 patients with prior myocardial infarction and residual inflammatory risk	Canakinumab 50, 150 or 300 mg, every 3 months	Placebo	Canakinumab compared to placebo resulted in: Significant decrease in hsCRP levels across all doses Significant decrease in the risk for the primary composite cardiovascular endpoint by 15% compared to placebo with canakinumab 150 mg * Cardiovascular efficacy was not proven for canakinumab 50 mg and canakinumab 300 mg
Russell et al., 2019 [52]	Randomized controlled trial	12 months	38 patients with symptomatic peripheral artery disease	Canakinumab 150 mg once monthly	Placebo	Canakinumab compared to placebo resulted in: Significant decrease in inflammatory markers Significant improvement in exercise capacity and mobility No effect on superficial femoral artery atherosclerotic plaque regression
Choudhury et al., 2016 [56]	Randomized controlled trial	12 months	189 patients with atherosclerotic disease and either type 2 diabetes or impaired glucose tolerance	Canakinumab 150 mg once monthly	Placebo	Canakinumab compared to placebo resulted in: Significant decrease in systemic inflammation markers Significant decrease in lipoprotein (a) levels No effect on vascular function and structure No effect on glycemic control and insulin resistance

**Table 2 jcm-12-01302-t002:** Overview of randomized controlled trials assessing the cardiovascular effects of IL-6 antagonists.

Study	Type of Study	Treatment Duration	Study Population	Utilized IL-6 Antagonist	Comparator	Main Findings
Tocilizumab						
Giles et al., 2020 [60]	Randomized controlled trial	3.2 years	3080 patients with seropositive, active rheumatoid arthritis	Tocilizumab 8 mg/kg/month	Etanercept 50 mg/week	No statistical difference between tocilizumab and etanercept across a number of surrogate cardiovascular endpoints
McInnes et al., 2015 [61]	Randomized controlled trial	24 weeks	132 patients with seropositive rheumatoid arthritis	Tocilizumab 8 mg/kg/month + methotrexate	Placebo + methotrexate	Tocilizumab compared to placebo resulted in: Significant increase in total cholesterol, low-density lipoprotein cholesterol (LDL-C) and triglycerides levels No significant effect on high-density lipoprotein cholesterol (HDL-C) and small, dense LDL Significant decrease in systemic inflammation markers No significant effect on arterial stiffness
Holte et al., 2017 [71]	Randomized controlled trial	Single dose of tocilizumab or placebo prior to coronary angiography	117 patients with non-ST segment elevation myocardial infarction	Tocilizumab 280 mg at a single dose	Placebo	Tocilizumab compared to placebo resulted in a: Non-significant effect on coronary flow reserve during hospitalization and 6 months post-discharge
Broch et al., 2021 [72]	Randomized controlled trial	Single dose of tocilizumab or placebo prior to coronary angiography	199 patients with ST-segment elevation myocardial infarction	Tocilizumab 280 mg at a single dose	Placebo	Tocilizumab compared to placebo resulted in a: Significant increase in myocardial salvage index Non-significant effect on final myocardial infarct size
Meyer et al., 2021 [73]	Randomized controlled trial	Single dose of tocilizumab	80 comatose with out-of-hospital cardiac arrest	Tocilizumab 8 mg/kg at a single dose	Placebo	Tocilizumab compared to placebo resulted in a: Significant decrease in systemic inflammation markers Significant decrease in myocardial injury and myocardial stress markers No significant effect on surrogate cardiovascular outcomes Significant increase in the risk for initiation of renal replacement therapy during intensive care unit stay
Ziltivekimab						
Ridker et al., 2021 [75]	Randomized controlled trial	24 weeks	66 patients with moderate to severe chronic kidney disease and residual inflammatory risk	Ziltivekimab 7.5 mg, 15 mg or 30 mg every 4 weeks	Placebo	Ziltivekimab compared to placebo resulted in a: Significant decrease in high-sensitivity CRP levels in a dose-dependent manner Significant decrease in other systemic inflammation markers

## Data Availability

Not applicable.

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
