# Peer review of "Do Interleukin-1 and Interleukin-6 Antagonists Hold Any Place in the Treatment of Atherosclerotic Cardiovascular Disease and Related Co-Morbidities? An Overview of Available Clinical Evidence"

_jcm, 2023, doi:10.3390/jcm12041302_

Round 1

Reviewer 1 Report

In the section 2 “inflammation & cardiovascular disease”, in both 2.1 and 2.2, while 2.1.1 and 2.2.1 focus on one specific CVD (atherosclerosis), some of the remaining subsections switched to the processes of CVDs, not on other types of CVDs. This needs fixed. Please review the roles of IL1b and IL6 in CVDs.

Figure 1 legends should be expanded and more detail information should be provided. Figure 1 was also not mentioned in the text.

The organization of the review could be improved by using section 3 “IL1 and IL6 antagonists for the treatment of CVD” and divided section 3 to 3.1 “IL1 antagonist” and 3.2 “IL6 antagonist”.

Also, in the new 3.1 and 3.2, the role of IL1 and IL6 antagonist should start with the name of CDVs that the IL1 or IL6 antagonist is intended to. Like  addition of “MI” before “A previous, pilot randomized controlled trial (RCT) enrolling 10 patients with ST-183 segment elevation MI (STEMI), who were”, see line 183

It is puzzling why the role of IL33 was discussed in 2.1.2

Author Response

Reviewer #1

In the section 2 “inflammation & cardiovascular disease”, in both 2.1 and 2.2, while 2.1.1 and 2.2.1 focus on one specific CVD (atherosclerosis), some of the remaining subsections switched to the processes of CVDs, not on other types of CVDs. This needs fixed. Please review the roles of IL1b and IL6 in CVDs.

Answer: We cordially thank the reviewer for this important comment. Indeed, the sub-sections entitled “Cardiac remodeling” rather represent a process and not a form of CVD; therefore, those sub-sections have been removed from the revised manuscript.

Figure 1 legends should be expanded and more detail information should be provided. Figure 1 was also not mentioned in the text.

Answer: We thank the reviewer for this important comment. Figure 1 legend has been expanded and figure 1 is cited in the revised manuscript.

The organization of the review could be improved by using section 3 “IL1 and IL6 antagonists for the treatment of CVD” and divided section 3 to 3.1 “IL1 antagonist” and 3.2 “IL6 antagonist”.

Answer: We thank the reviewer for this comment. Section 3 of the manuscript has been re-arranged according to reviewer’s suggestion.

Also, in the new 3.1 and 3.2, the role of IL1 and IL6 antagonist should start with the name of CDVs that the IL1 or IL6 antagonist is intended to. Like  addition of “MI” before “A previous, pilot randomized controlled trial (RCT) enrolling 10 patients with ST-183 segment elevation MI (STEMI), who were”, see line 183

Answer: We cordially thank the reviewer for this significant comment. We preserved the headlines according to the specific IL-1 or IL-6 antagonist, while we added headlines according to the specific form of CVD that each drug is intended to (except for data from studies from primary prevention cohorts).

It is puzzling why the role of IL33 was discussed in 2.1.2

Answer: We thank the reviewer for this important and interesting comment. IL-33 was discussed in the corresponding section of the manuscript, since it has been recognized as a core member of the IL-1 family (Cayrol C, Girard JP. Interleukin-33 (IL-33): A nuclear cytokine from the IL-1 family. Immunol Rev. 2018 Jan;281(1):154-168. doi: 10.1111/imr.12619. PMID: 29247993, & Chen WY, Tsai TH, Yang JL, Li LC. Therapeutic Strategies for Targeting IL-33/ST2 Signalling for the Treatment of Inflammatory Diseases. Cell Physiol Biochem. 2018;49(1):349-358. doi: 10.1159/000492885. Epub 2018 Aug 23. PMID: 30138941.). Corresponding section has been expanded, in order to be easier for the reader to understand the interconnection between IL-1 and IL-33.

Reviewer 2 Report

The review by Dimosiari et al summarizes the data regarding the role of interleukin-1 and -6 in cardiovascular diseases and summarizes available data from clinical trials that used anti-interleukin-1/6 therapies. The review is well-written and organized; however, I believe there are issues to be re-addressed.

Major

1.        Tables 1 and 2 contain mainly the same information as the text under paragraphs 3 and 4, respectively. Moreover, in my opinion, both tables are not reader-friendly because i) the Main Findings column is narrow while contains much text; ii) there is no horizontal grids to separate individual studies. I would rather suggest reformatting the tables with Improvement, No-effect, and Reference columns while identifying the main findings (theses) as short as possible. Alternative presentation may be used (it is up to the authors) but the main requirement is conciseness.

2.        Additionally, despite the available studies used different agents, these agents were directed against either IL1 or IL6. Therefore, the review would gain value if common promising trends (potential benefits to patients) across the studies are identified as a summary conclusion at the end of Paragraphs 3 and 4.

Minor

1.        The text in lines 443-445 starting with ‘6. Patients…’ has to be deleted.

Author Response

Reviewer #2

The review by Dimosiari et al summarizes the data regarding the role of interleukin-1 and -6 in cardiovascular diseases and summarizes available data from clinical trials that used anti-interleukin-1/6 therapies. The review is well-written and organized; however, I believe there are issues to be re-addressed.

Major

  1. Tables 1 and 2 contain mainly the same information as the text under paragraphs 3 and 4, respectively. Moreover, in my opinion, both tables are not reader-friendly because i) the Main Findings column is narrow while contains much text; ii) there is no horizontal grids to separate individual studies. I would rather suggest reformatting the tables with Improvement, No-effect, and Reference columns while identifying the main findings (theses) as short as possible. Alternative presentation may be used (it is up to the authors) but the main requirement is conciseness.

Answer: We cordially thank the reviewer for this targeted comment. Indeed, eligible studies are very heterogeneous in terms of assessed outcomes and endpoints of interest and therefore it is really difficult to be summarized in the format suggested. We re-formatted the corresponding tables, in order to make last column with each study’s main findings more “reader friendly”.

  1. Additionally, despite the available studies used different agents, these agents were directed against either IL1 or IL6. Therefore, the review would gain value if common promising trends (potential benefits to patients) across the studies are identified as a summary conclusion at the end of Paragraphs 3 and 4.

Answer: We heartily thank the reviewer for this very important comment. According to the reviewer’s suggestion, a concluding remark for each IL-1 or IL-6 antagonist has been added at the end of each corresponding section.

Minor

  1. The text in lines 443-445 starting with ‘6. Patients…’ has to be deleted.

Answer: We thank the reviewer for this comment. Corresponding text has been deleted from the revised version of our manuscript.

Round 2

Reviewer 1 Report

The revised version has been improved. Please make following minor changes.

1. no 2.1.3 and 2.2.3

2. please add "active" before IL1b (in line 70), to indicate that IL1b is activated.  (resulting in formation of "active" interleukin-1β (IL-1β) .

3. Paragraph in lines 191-199 should be moved after "3.1 IL1 antagonists"

4. please removing the space before "Ankinra" in line 202.

Author Response

We would like to thank the reviewer for this minor changes.

All comments have been addressed and relevant modification has been made in the revised manuscript.

Thank you once again for contributing to the improvement of our manuscript's overall quality.